# Factors associated with students' unwillingness to voluntarily donate blood in the Eastern Region of Ghana

Foster Amponsah-Manu[1], Augustine Badu[1,2]*, Yubraj Acharya[3], Augustine kyere[1], Philomena Quayson[1], Jocelyn Asibey[1], Samuel Attua[1], Richmond Afrifa[1], Eunice Serwaa Kegyeda[1], Samuel Morton[1], Richmond Ampofo[1], Gloria Akofa Quarshie[1], Offeibea-Effah Adarkwa[1], Arko Akoto Ampaw[1]

1 Department of Surgery, Eastern Regional Hospital, Koforidua, Eastern Region, Ghana, 2 Faculty of Health and Allied Sciences, Koforidua Technical University, Koforidua, Eastern Region, Ghana, 3 Department of Health Policy and Administration, The Pennsylvania State University, University Park, PA, United States of America

* augustine.badu@ktu.edu.gh

## Abstract

### Background

The World Health Organisation advocates for voluntary blood donation (BD) as a sustainable means of maintaining adequate blood availability in healthcare facilities globally. However, the Eastern Region of Ghana experiences low voluntary BD rates among students. The study aimed to determine factors associated with this pattern, conceptualising unwillingness as a key determinant of low voluntary BD.

### Method

The analytical cross-sectional study was conducted from July to August 2025. Respondents were engaged with a structured questionnaire which gathered information on demographic characteristics, prior donation experience, knowledge about blood donation, sociocultural influences, institutional and logistical factors. Descriptive, Chi-square and multivariable regression analyses were done with Stata® version 15.0. Statistical significance was assumed at P-values <0.05.

### Results

A total of 2,200 students participated in the study, and 40.3% (887/2,200) of them were unwilling to voluntarily donate blood (VDB). Unwillingness was significantly associated with perceived ineligibility for blood donation (aOR 3.45, 95% CI: 2.63–4.52), belief that blood donation is unsafe (aOR 1.98, 95% CI: 1.56–2.49), low knowledge about blood donation (aOR 1.68, 95% CI: 1.05–2.69), and discouragement by cultural or religious background (aOR 1.85, 95% CI: 1.25–2.73). Institutional and logistical barriers including long waiting times (aOR 4.31, 95% CI: 2.14–8.70), lack of

**Data availability statement:** All relevant data are within the paper and its Supporting Information file.

**Funding:** The author(s) received no specific funding for this work.

**Competing interests:** The authors have declared that no competing interests exist.

incentives (aOR 3.85, 95% CI: 1.85–8.02) and inconvenient donation locations (aOR 3.04, 95% CI: 1.51–6.09), were also independently associated with unwillingness. Free health check-ups were the most preferred incentive.

## Conclusion

The study revealed important factors influencing students' unwillingness to VDB. Addressing misconceptions, improving education on eligibility and safety, and reducing logistical barriers may enhance students' willingness and potentially increase voluntary BD rate.

---

## 1. Background

Blood donation is a critical component of healthcare systems worldwide, providing life-saving transfusions for patients in need due to surgeries, trauma, chronic illnesses, and complications related to pregnancy and childbirth. Blood donors often fall into three categories: paid or commercial donors, family/replacement donors, and volunteers [1]. According to WHO, all nations should implement a voluntary non-remunerated blood donation strategy and ensure that all blood products are sustainably provided by unpaid volunteers [1]. About 118 million blood donations are made worldwide annually, with high-income nations spearheading the effort and accounting for 40% of all donations [2]. In Ghana, despite the national required target of 344,000 units of blood in 2024, the National Blood Service collected 187,280 units representing 54.4% of the required units [3]. Not only is Ghana not meeting its target for donated blood, but voluntary blood donation rate as a share of total donations also remains low. Only 29% of blood donations collected in 2024 were voluntary [3], indicating continued reliance on family replacement and commercial donors.

Tertiary and second-cycle students represent a demographic group with the potential to contribute significantly to blood supply. Tertiary students are university, polytechnic and college of education students. Whereas second-cycle students comprise of students in post-basic educational institutions (senior high, technical, vocational, business and agricultural institutions) [4].

They are generally young, healthy, and capable of donating blood, often through organised blood donation drives. However, voluntary blood donation rates among students remain suboptimal in many settings [5–7], despite their potential. The Eastern Region of Ghana experiences similar pattern of low donation rates, according to records of the Blood Donation Unit (BDU) of the Eastern Regional Hospital (ERH). The BDU recorded an average of 15.7% donation rate across schools visited in 2024. Understanding individual-level factors associated with this pattern is crucial for designing interventions to encourage voluntary donation among students in the region.

Blood donation behaviour is determined by a series of psychological and structural factors [8]. Research and literature from behavioural sciences highlight the difference between intention (willingness) and actual donation behaviour [8,9]. This behavioural

difference is confirmed by a structural equation modelling study, which found that intent to donate blood did not significantly load on the model that predicted the actual behaviour to donate blood [9]. However, a systematic review found that within extended behavioural models, intentions (or willingness) are associated with subsequent donation behaviour [8]. This indicates that willingness is an important psychological determinant within voluntary blood donation pathway. Studies have reported unwillingness to limit the participation of students in voluntary blood donation [7,10]. Thus, this study conceptualises unwillingness as a key determinant of low voluntary blood donation rate.

Previous studies have identified some factors that influence students' willingness or intention to donate blood. A study in China reported attitude towards blood donation (with willingness as a key component) as a mediating effect through which sociodemographic characteristics, health status and knowledge about blood donation, affected blood donation [11]. In Serbia, fear of needles, lack of motivation and fear of adverse reactions after donating blood were key deterrents, with donor experience and educational level influencing positive attitude towards blood donation [12]. Additionally, cultural beliefs and religious perceptions have been suggested to play a significant role in shaping attitudes towards blood donation in Sub-Saharan Africa. Some individuals hold superstitious beliefs that donating blood can weaken the body or expose them to spiritual harm, leading to unwillingness to donate blood [13].

Another reported hindrance to willingness to donate is inadequate knowledge about blood donation. More than half of student respondents reported not knowing the amount of blood donated at a time in Tanzania [14]. Lack of knowledge may encourage misconceptions and fear about the donation process, which may reduce individuals' willingness to donate, even when opportunities exist.

Institutional and logistical factors may also influence students' willingness to donate blood voluntarily. Inconvenient donation locations, long waiting times, lack of perceived incentives, lack of health promotion platforms in schools, limited exposure to blood donation education, academic schedules and lack of transportation have been reported as barriers in resource-limited settings [5,15]. These factors are often reported in relation to turnout, but they may shape willingness by influencing perceptions of convenience, safety, and benefits that one may obtain.

Many of the studies on students' blood donation practices have been conducted outside Ghana, and there is limited information specifically on students' unwillingness to voluntarily donate blood in the Eastern Region. We therefore investigated unwillingness as a distinct outcome representing a psychological barrier to voluntary blood donation among students in the Region.

## 2. Methods

### 2.1 Study area

The study was conducted in tertiary and second-cycle schools in the Eastern Region of Ghana. The region is one of the 16 regions of Ghana, with its capital being Koforidua. It is bordered to the east by the Lake Volta, to the north by Bono East and Ashanti Regions, to the west by Ashanti Region, and to the south by Central and Greater Accra Regions [16]. The region, through the local government administrative systems, is divided into 13 municipalities and 20 districts [16]. The region is home to several universities, colleges, senior high and technical schools, making it a suitable location for the study.

### 2.2 Research design and population

A cross-sectional analytical research design was adopted to investigate the factors associated with students' unwillingness to voluntarily donate blood (VDB) in the Eastern Region of Ghana from July to August 2025. The study was conducted by the Blood Donation Committee of the ERH, led by the Clinical and Medical Directors.

The target population for the study consisted of students enrolled in selected 21 second-cycle and 3 tertiary institutions in the Region during the duration of the study.

## 2.3 Sample size estimation

The primary outcome of the study was willingness to donate blood (binary; =1 if the respondent said they would be willing to donate blood). Although we intended to examine several key determinants, our sample size calculation was based on the students' knowledge about blood donation (BD), as this would provide a potential entry point for intervention (for example, if we found that students with limited knowledge were less likely to donate blood, the intervention would be educational programs to increase that knowledge). Based on data from the National Blood Service of Ghana, we assumed that approximately 30% students would be willing to donate blood. We also assumed that there would be three types of students: those with very little knowledge, those with some knowledge, and those with a lot of knowledge. They correspond to Low, Moderate, and High in "Knowledge about BD" measure in Table 2. We assumed that the proportion of students in these categories would be roughly 20%, 60%, and 20%. We also assumed about a 10 percentage point difference in the probability of being willing to donate blood (equivalent to an odds ratio of about 1.55) between High and Low would be meaningful. Under these assumptions, a sample size of 1,920 students—or, approximately, 80 students each from the 24 schools—provides approximately 90% power (on a one-sided test with significance level of 0.05). When we visited the schools for data collection, we surveyed all eligible students since the marginal cost of collecting data from an additional student once we were there was low.

## 2.4 Selection of schools

The Eastern Region has a total of 74 second-cycle and 31 tertiary schools [17,18]. Out of these, the BDU of the ERH visited 59 second-cycle schools in 15 districts and 6 municipalities, for voluntary BD drives in 2024. A proportionate stratified sampling technique was used to select 21 second-cycle schools from the 59 schools visited.

First, the number of schools in each district or municipality was determined, and the proportion of schools in each district or municipality relative to the total (59) was calculated. Based on these proportions, the number of schools to be selected from each district out of the required sample of 21 was determined. Values of 0.5 and above were rounded up to the nearest whole number, while values below 0.5 were rounded down.

After determining the number of schools to be selected from each district, the specific schools were chosen through simple random sampling (balloting). Districts or municipalities with 7, 5, 4, 3, 2, and 1 schools contributed 2, 2, 1, 1, 1, and 0 schools, respectively. This process initially resulted in 20 schools. Therefore, one additional school was randomly selected from a district that had one school but no initial selection, bringing the final sample to 21 schools.

Also, the BDU visited 5 tertiary schools in 5 different districts for voluntary BD drives in 2024. Three tertiary schools were randomly selected (balloted) out of the 5 visited. Thus, the study included 21 and 3 randomly selected second-cycle and tertiary schools respectively.

## 2.5 Recruitment of respondents

A systematic random sampling technique was adopted for the recruitment of students. In each selected school, the sampling frame consisted of the student register obtained from the school administration. Eligible students were those present at the time of data collection and met the inclusion criteria. A sampling interval (K) was calculated by dividing the student population by the allocated sample size for each school (n = 80). The first respondent was sampled by simple random, then every 'K$^{-th}$' student was sampled until the required sample size was reached or exceeded.

## 2.6 Inclusion and exclusion criteria

The study included students 17 years and above in the selected tertiary and second-cycle schools in the Eastern Region, who willingly provided written informed consent. Respondents who were unable to complete the study questionnaire for whatever reason were excluded from the study.

## 2.7 Data collection

Data were collected using a structured questionnaire. The questionnaire included both close-ended and Likert-scale questions and covered the following sections: demographic factors (age, gender, educational level, year of study and residential status and religious affiliation); knowledge and experience of BD (donation history, perceived BD eligibility, information received on BD, safety of BD process and assessment of knowledge about BD); sociocultural factors (peer and family influence, influence by culture/religion, faces social stigma and belief in myths about BD); institutional and logistical factors (frequency of BD drives, awareness of BD drives, BD education at student orientation programmes, health club promoting BD, location of BD drives, incentives, waiting times, safety concerns and notification of event date). The questionnaire was pretested with 50 second-cycle students from a non-participating school and ambiguities as well as sensitive questions were addressed. Respondents were handed the questionnaires to fill out after the purpose of the survey had been explained to them in plain language and had signed the informed consent form.

Respondents who answered yes to the question 'Would you be willing to voluntarily donate blood in future?' were classified as willing to voluntarily donate blood. Those who answered no or not sure to the same question were classified as unwilling to voluntarily donate blood (VDB).

Belief in myths about BD was assessed by asking in the questionnaire whether respondents believed in some common myths about BD (BD makes you weak permanently; BD can make a man impotent and a woman barren; BD reduces life expectancy (can make you die early); BD can make you a dull student; BD is linked to spiritual beliefs). Those who answered yes to any of the myths, were deemed to believe in myths about BD.

Also, respondents were asked whether there was any social stigma or taboo to BD in their school or community. Those who answered yes to the question were deemed to face social stigma for BD.

## 2.8 Data analysis

Survey responses were analysed with Stata® 15.0 version. Descriptive statistics such as frequencies, percentages, means, and standard deviations were used to summarise data. Chi-square test and multivariable logistic regression analysis were performed to identify significant factors influencing students' unwillingness to voluntarily donate blood. Statistical significance was assumed at P-value <0.05. All the significant bivariate variables were entered into a multivariable analysis. Standard errors were clustered at the level of the schools (N = 24), thus allowing arbitrary correlation in the unwillingness to donate blood among students in a given school.

## 2.9 Ethical considerations

Ethical clearance was obtained from the Koforidua Technical University Ethical Review Committee with reference number KTU/DRI/ERB/GAI/12. Permission to conduct the study was obtained from the Eastern Regional Directorates of Education and Health, as well as the management of each school. Written informed consent was obtained from all respondents after the ethical statement, including the purpose and procedure of the study, as well as their right of withdrawal, was explained to them in plain English language. Written consent was obtained from head masters/mistresses for participants less than 18 years, who were primarily second-cycle students. The anonymity and confidentiality of respondents were maintained at all times as they were given identification numbers, and no personal identifiers were taken. Respondents were engaged for the study from 30th July 2025 to 22nd August 2025. Completed questionnaires and informed consent forms are stored securely at ERH.

## 3. Results

A total of 2,573 students from the 24 institutions were invited to participate in the study using systematic random sampling. Out of these, 2,467 students filled the questionnaires, yielding a response rate of 95.9%. However, 267 responses were excluded based on the study's exclusion criteria, resulting in a final sample of 2,200 respondents.

### 3.1 Demographic variables and willingness to VDB

The study recorded more female students (58.1%, 1278/2,200) than male students. The majority of the respondents were second-cycle students, Christians, and resided on campus. Close to half of the respondents (40.3%, 887/2,200) indicated unwillingness to VDB (Table 1). Fourth-year tertiary students were more likely not to VDB than 1st year tertiary and second-cycle students (aOR 1.83, 95% CI: 1.20–2.77) (Table 1).

### 3.2 Experience and knowledge about blood donation and willingness to VDB

Only 13.5% (298/2,200) of respondents had donated blood before and majority of them (79.5%, 237/298) were willing to VDB in future. Perceived ineligibility was strongly associated with unwillingness to VDB (aOR 3.45, 95% CI: 2.63–4.52) (Table 2), with young age as the most cited reason (46.3%, 326/ 704) (Fig 1). Close to half of the respondents who had never received information on BD (47.83%, $X^2$: 38.66, p < 0.010) were unwilling to VDB. Low Knowledge about BD (aOR 1.68, 95% CI: 1.05–2.69), belief of BD as an unsafe process (aOR 1.98, 95% CI: 1.56–2.49) were independently associated with unwillingness to VDB (Table 2). Risk of infection (59.3%, 211/ 356) and anaemia (40.7%, 145/356) were cited as the main reasons for belief of BD as unsafe.

**Table 1. Bivariate and multivariable analysis of demographic variables and willingness to voluntarily donate blood (VDB).**

| Demographic characteristics | | Willingness to VDB | | | Unwillingness to VDB |
|---|---|---|---|---|---|
| | | No / Not sure | Yes | | |
| | N = 2,200 | N = 887 (40.3%) | N = 1,313 (59.7%) | | |
| | n | n (%) | n (%) | $X^2$ (p-value) | aOR (95% CI) |
| Age in years (Mean ± SD) = 17.97 ± 1.6 | | | | | 1.04 (0.96, 1.12) |
| **Gender** | | | | | |
| Female | 1,278 | 511 (39.98) | 767 (60.02) | 0.14 | |
| Male | 922 | 376 (40.78) | 546 (59.22) | (0.707) | |
| **Educational Level** | | | | | |
| Tertiary | 253 | 66 (26.09) | 187 (73.91) | 24.06 | Ref |
| 2nd cycle | 1,947 | 821 (42.17) | 1,126 (57.83) | **(0.000)** | 1.27 (0.73, 2.21) |
| **Year of study** | | | | | |
| 1st | 191 | 55 (28.8) | 136 (71.2) | 13.99 | Ref |
| 2nd | 882 | 362 (41.04) | 520 (58.96) | **(0.003)** | 1.41 (0.85, 2.33) |
| 3rd | 1,087 | 458 (42.13) | 629 (57.87) | | 1.42 (0.82, 2.43) |
| 4th | 40 | 12 (30.00) | 28 (70.00) | | **1.83 (1.20, 2.77) \*\*** |
| **Residential status** | | | | | |
| Off-campus | 448 | 189 (42.19) | 259 (57.81) | 0.82 | |
| On-campus | 1,752 | 698 (39.84) | 1,054 (60.16) | (0.366) | |
| **Religious affiliation** | | | | | |
| Christianity | 2,001 | 809 (40.43) | 1,192 (59.57) | 4.19 | |
| Islam | 178 | 67 (37.64) | 111 (62.36) | (0.242) | |
| Traditional | 12 | 8 (66.67) | 4 (33.33) | | |
| Others | 9 | 3 (33.33) | 6 (66.67) | | |

Note: P-value: * < 0.05, ** < 0.010, *** < 0.001; $X^2$ - Pearson Chi-square statistic; VDB- voluntarily donate blood; percentages across rows sum up to 100%; Significant values are bolded.

**Table 2. Bivariate and multivariable analysis of experience and Knowledge about blood donation and willingness to VDB.**

| Experience and Knowledge about BD | | Willingness to VDB | | | Unwillingness to VDB |
|---|---|---|---|---|---|
| | | No / Not Sure | Yes | | |
| | n | n (%) | n (%) | X² (p-value) | aOR (95% CI) |
| **Donated blood before?** | | | | | |
| No | 1,902 | 826 (43.43) | 1076 (56.57) | 56.43 | 1.35 (0.96, 1.87) |
| Yes | 298 | 61 (20.47) | 237 (79.53) | (0.000) | Ref |
| **Perceived BD eligibility** | | | | | |
| No | 704 | 369 (52.41) | 335 (47.59) | 205.16 | **3.45 (2.63, 4.52) \*\*\*** |
| Not sure | 719 | 362 (50.35) | 357 (49.65) | (0.000) | **2.64 (1.89, 3.68) \*\*\*** |
| Yes | 777 | 156 (20.08) | 621 (79.92) | | Ref |
| **Ever received information on BD?** | | | | | |
| No | 943 | 451 (47.83) | 492 (52.17) | 38.66 | 1.18 (0.948, 1.48) |
| Yes | 1,257 | 436 (34.69) | 821 (65.31) | (0.000) | Ref |
| **Belief of BD as safe process** | | | | | |
| No | 356 | 188 (52.81) | 168 (47.19) | 144.54 | **1.98 (1.56, 2.49) \*\*\*** |
| Not sure | 659 | 359 (54.48) | 300 (45.52) | (0.000) | **1.87 (1.47, 2.37) \*\*\*** |
| Yes | 1,185 | 340 (28.69) | 845 (71.31) | | Ref |
| **Knowledge about BD** | | | | | |
| Low | 233 | 154 (66.09) | 79 (33.91) | 83.14 | **1.68 (1.05, 2.69) \*** |
| Moderate | 1,660 | 645 (38.86) | 1,015 (61.14) | (0.000) | 1.05 (0.68, 1.62) |
| High | 307 | 88 (28.66) | 219 (71.34) | | Ref |

Note: P-value: \*<0.05, \*\*<0.010, \*\*\*<0.001; *X²* - Pearson Chi-square statistic; VDB- voluntarily donate blood, BD- blood donation; percentages across rows sum up to 100%; Significant values are bolded.

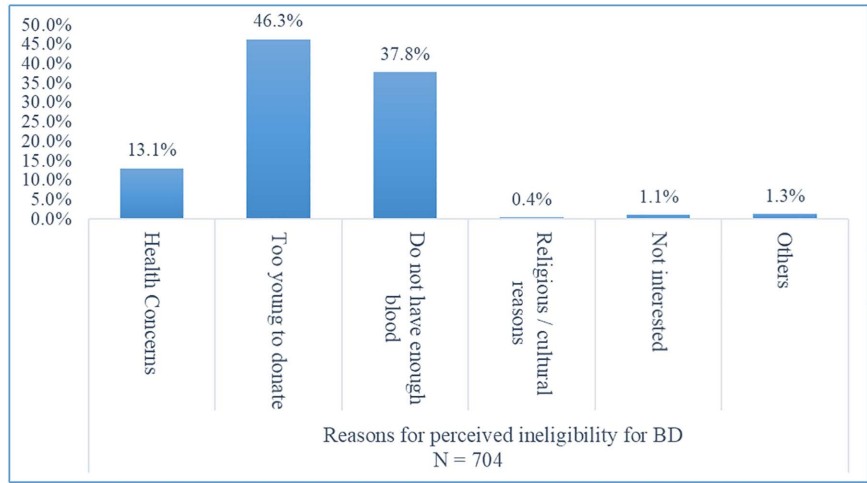

**Fig 1. Respondents' distribution on reasons for perceived ineligibility for BD.** Note: BD = blood donation.

### 3.3 Sociocultural variables and willingness to VDB

A higher proportion of respondents whose peers did not donate blood (48.8%, $X^2$: 52.04, p<0.010), had never been encouraged by family/friends to donate blood (47.48%, $X^2$: 55.52, p<0.010), faced social stigma to blood donation (43.7%, $X^2$: 38.12, p<0.05), believed in myths about BD (38.8%, $X^2$: 24.06, p<0.05), were unwilling to VDB (Table 3). Discouragement of BD by cultural/religious background was independently associated with unwillingness to VDB on multivariable analysis (aOR 1.85, 95% CI: 1.25–2.73) (Table 3).

### 3.4 Institutional/logistical variables and willingness to VDB

A higher proportion of respondents whose institutions rarely organised BD drives (48.35%, $X^2$: 27.65, p<0.010), were not aware of the most recent BD drive (43.92%, $X^2$: 9.08, p<0.05), had no health club that promotes BD (41.95%, $X^2$: 7.33, p<0.05), were unwilling to VDB (Table 4). Students' orientation not including education on BD had a weak positive association with unwillingness to VDB (aOR 1.23, 95% CI: 1.02–1.48). Logistical challenges including inconvenient location (aOR 3.04, 95% CI: 1.51–6.09), lack of incentives (aOR 3.85, 95% CI: 1.85–8.02), long waiting times (aOR 4.31, 95% CI: 2.14–8.70), safety concerns (aOR 4.46, 95% CI: 2.08–9.59) and poor notification of event date (aOR 2.58, 95% CI: 1.30–5.12), were strongly associated with unwillingness to VDB (Table 4). The most preferred incentive was free health checkups (72.1%, 1,149/ 1,594) (Fig 2).

**Table 3. Bivariate and multivariable analysis of sociocultural variables on blood donation and willingness to VDB.**

| Sociocultural variables | | Willingness to VDB | | | Unwillingness to VDB |
|---|---|---|---|---|---|
| | n | No/ Not sure | Yes | $X^2$ (p-value) | |
| | | n (%) | n (%) | | aOR (95% CI) |
| **Peers donate blood?** | | | | | |
| Yes | 1,095 | 359 (32.79) | 736 (67.21) | 52.04 | Ref |
| Not sure | 478 | 222 (46.44) | 256 (53.56) | **(0.000)** | 1.29 (0.99, 1.67) |
| No | 627 | 306 (48.8) | 321 (51.20) | | 1.14 (0.91, 1.45) |
| **Encouraged by family /peers to donate blood?** | | | | | |
| Yes | 982 | 313 (31.87) | 669 (68.13) | 55.52 | Ref |
| Not sure | 26 | 8 (30.77) | 18 (69.23) | **(0.000)** | 0.52 (0.23, 1.15) |
| No | 1,192 | 566 (47.48) | 626 (52.52) | | 1.21 (0.99, 1.46) |
| **Culture/ religious background discourages BD?** | | | | | |
| No | 1,472 | 496 (33.70) | 976 (66.30) | 81.23 | Ref |
| Not sure | 352 | 192 (54.55) | 160 (45.45) | **(0.000)** | **1.55 (1.14, 2.11) \*\*** |
| Yes | 376 | 199 (52.93) | 177 (47.07) | | **1.85 (1.25, 2.73) \*\*** |
| **Faces social stigma to BD** | | | | | |
| No | 1,639 | 602 (36.73) | 1,037(63.27) | 38.12 | Ref |
| Not sure | 426 | 226 (53.05) | 200 (46.95) | **(0.000)** | 1.04 (0.85, 1.29) |
| Yes | 135 | 59 (43.70) | 76 (56.30) | | 0.94 (0.56, 1.59) |
| **Belief in myths about BD** | | | | | |
| No | 1,310 | 486 (37.10) | 824 (62.90) | 24.06 | Ref |
| Not sure | 524 | 259 (49.43) | 265 (50.57) | **(0.000)** | 1.23 (0.88, 1.71) |
| Yes | 366 | 142 (38.80) | 224 (61.20) | | 0.94 (0.74, 1.19) |

Note: P-value: \*<0.05, \*\*<0.010, \*\*\*<0.001; $X^2$ - Pearson Chi-square statistic; VDB- voluntarily donate blood; BD- blood donation; percentages across rows sum up to 100%; Significant values are bolded.

**Table 4. Bivariate and multivariable analysis of institutional and logistic variables and willingness to VDB.**

| Institutional and logistic variables | | Willingness to VDB | | | Unwillingness to VDB* |
|---|---|---|---|---|---|
| | | No/ Not sure | Yes | | |
| | n | n (%) | n (%) | X² (p-value) | aOR (95% CI) |
| **Frequency of BD drives per year** | | | | | |
| > twice | 282 | 106 (37.59) | 176 (62.41) | 27.65 | Ref |
| Once or twice | 1,219 | 443 (36.34) | 776 (63.66) | (0.000) | 0.89 (0.66, 1.19) |
| Rarely | 699 | 338 (48.35) | 361 (51.65) | | 1.09 (0.83, 1.42) |
| **Aware of the most recent BD drive?** | | | | | |
| Yes | 1,246 | 468 (37.56) | 778 (62.44) | 9.08 | Ref |
| No | 954 | 419 (43.92) | 535 (56.08) | (0.003) | 1.01 (0.82, 1.25) |
| **Student orientation programmes includes education on BD?** | | | | | |
| Yes | 581 | 180 (30.98) | 401 (69.02) | 30.33 | Ref |
| Not sure | 549 | 252 (45.90) | 297 (54.10) | (0.000) | **1.28 (1.04, 1.57) *** |
| No | 1,070 | 455 (42.52) | 615 (57.48) | | **1.23 (1.02, 1.48) *** |
| **Health club promoting BD?** | | | | | |
| Yes | 548 | 194 (35.40) | 354 (64.60) | 7.33 | Ref |
| No | 1,652 | 693 (41.95) | 959 (58.05) | (0.007) | 0.98 (0.76, 1.27) |
| **Logistical challenges discouraging BD** | | | | | |
| Others | 50 | 6 (12.00) | 44 (88.00) | 42.27 | Ref |
| Inconvenient location | 143 | 45 (31.47) | 98 (68.53) | (0.000) | **3.04 (1.51, 6.09) ** ** |
| Lack of incentives | 487 | 199 (40.86) | 288 (59.14) | | **3.85 (1.85, 8.02) *** ** |
| Long waiting times | 709 | 291 (41.04) | 418 (58.96) | | **4.31 (2.14, 8.70) *** ** |
| Safety concerns | 601 | 283 (47.09) | 318 (52.91) | | **4.46 (2.08, 9.59) *** ** |
| Poor notification of event date | 210 | 63 (30.00) | 147 (70.00) | | **2.58 (1.30, 5.12) ** ** |

Note: P-value: *<0.05, **<0.010, ***<0.001; X² - Pearson Chi-square statistic; VDB- voluntarily donate blood; BD- blood donation; percentages across rows sum up to 100%; Significant values are bolded.

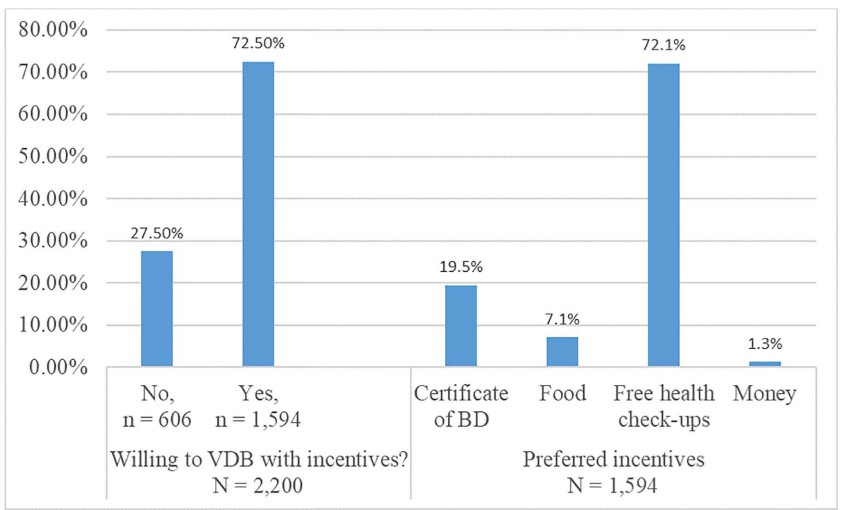

**Fig 2. Respondents' distribution on willingness to VDB with incentives and preferred incentives.** Note: VDB- voluntarily donate blood, BD- blood donation.

## 4. Discussion

The study investigated factors influencing students' unwillingness to voluntarily donate blood (VDB) in the Eastern Region of Ghana. The study provides insight into individual-level barriers that may contribute to low voluntary blood donation (BD), by assessing unwillingness as a distinct behavioural outcome. A substantial proportion of students (40.3%, 887/2,200) were unwilling to VDB, representing a worrying hesitant donor population. Unwillingness was largely influenced by cognitive, sociocultural, and structural (institutional and logistical) factors.

The findings of the study entrench the importance of education in improving voluntary BD by influencing willingness. Low knowledge and lack of information were positively associated with unwillingness to VDB. Lack of information was one of the major reasons for not donating blood among university students in Kenya [19]. In Ethiopia, participants with adequate Knowledge about BD were more likely to donate blood [20]. Knowledge alone may not guarantee willingness. However, insufficient knowledge appears to hinder willingness to donate. Educational interventions integrated into student orientation programmes or institutional health promotion initiatives may play a crucial role in enhancing willingness to VDB.

On blood donation experience, the study recorded a low voluntary blood donation rate (13.5%). However, a higher proportion (79.5%) of those who had donated before were willing to donate in future. This highlights the importance of positive experiences in shaping willingness to donate again. The voluntary BD rate recorded by the study is comparable to rates reported in Ethiopia (12.4), Portugal (12.7%) and Nigeria (15%) [20–22]. It is, however, lower than the rate (31.7%) reported among tertiary students at the University of Cape Coast in Ghana [6]. The low rate is consistent with the records of the Eastern Regional Hospital, which recorded an average of 15.7% across schools visited in 2024. This underscores the urgency for remedial measures to enhance willingness for potential improvement in donation rate.

Perceived ineligibility emerged as one of the strongest independent predictors of unwillingness. Notably, many students cited young age as a reason for ineligibility, despite meeting the minimum age requirement. Perceived ineligibility was reported as one of the barriers to BD among university students in Saudi Arabia [23]. A study conducted in the Northern Region of Ghana revealed that only about 36.0% of respondents knew the minimum age for BD [24]. These findings suggest a knowledge gap regarding donation criteria. Thus, there is a need to clarify eligibility requirements during health education campaigns to reduce misconceptions about eligibility. Misconceptions about eligibility may function as a cognitive barrier that hinders intention to VDB.

The belief of blood donation as an unsafe process was also strongly associated with unwillingness to VDB, with risk of infection and anaemia reported as the main reasons. Fear of anaemia was a key barrier to BD in Ethiopia [25]. Post BD adverse effects like dizziness, tiredness and feeling weak were identified as barriers to BD in Kenya and Ghana [6,19]. The fear of contracting infectious diseases was one of the main barriers to BD in Saudi Arabia [23]. These fears represent an important psychological barrier that hinders intention formation. Education about safety protocols and post-donation recovery may reduce these concerns and improve willingness.

On sociocultural factors, the results highlight the positive influence of peers and family on willingness to VDB. Family history of BD was associated with the practice of BD in a study by Dejene et al. (2021). In the Northern Region of Ghana, respondents were more likely to donate blood if their peers or role models actively participated [24]. These findings affirm that observational learning and normative reinforcement shape behavioural intentions. Thus, creating peer networks and student ambassadors may present a good strategy to enhance willingness to VDB.

Belonging to a particular religion *per se* had no significant association with willingness to VDB, but discouragement of BD by cultural/religious background had a positive association with unwillingness to VDB. This finding suggests that interpretive beliefs and social norms may influence unwillingness to VDB, but not formal affiliation. Some studies have reported no association between religion and BD practices [25,26]. In Saudi Arabia, religion was identified as one of the motivational factors for BD [23]. This suggests that identifying and consciously addressing misconceptions and aspects of religion and culture that discourage BD may improve intention or willingness.

Institutional and logistical factors were also associated with unwillingness to VDB. A higher proportion of respondents whose institutions rarely organised BD drives, were not aware of the most recent BD drive, whose student orientation programmes did not include education on BD, had no health club/association that promotes BD, were unwilling to VDB. Studies have reported lack of health promotional platforms in schools and limited exposure to BD education as barriers in resource-limited settings [6,15]. These findings emphasize that lack of institutional efforts in terms of education on BD and the provision of enabling health promotion platforms hinders intention formation towards voluntary BD.

Long waiting times, inconvenient locations, safety concerns, poor notification of donation events, and lack of incentives were all independently and strongly associated with unwillingness. These findings are consistent with a scoping review in sub-Saharan Africa and studies in Ghana [6,15,24]. These factors are often discussed in relation to turnout, but the findings suggest they also influence willingness. Students may form negative intentions and attitudes, when donation processes are perceived as inconvenient, unsafe or unrewarding.

The preference for free health check-ups as an incentive is consistent with other studies [25,27]. Non-remunerated BD is highly recommended in line with WHO's recommendations. However, non-monetary incentives such as health screening may enhance willingness without compromising ethical standards.

Demographic characteristics were largely not associated with unwillingness. Studies conducted in Ethiopia and Saudi Arabia have reported conflicting findings about the influence of gender on intention to practice BD [20,23,28]. These findings suggest gender-based geographical influence on intentions to VDB, underpinned by possible socio-cultural differences.

Although age was not associated with unwillingness, fourth-year tertiary students were more likely to report unwillingness compared to first-year tertiary and second-cycle students. This may be attributed to competing academic priorities and additional responsibilities, reducing their willingness to engage in extracurricular activities, including voluntary BD. Our findings suggest that interventions may yield greater impact if targeted toward first years.

A key limitation of the study is its cross-sectional nature, which limits a causal interpretation of the estimated coefficients. Another limitation is that the findings may not apply to other regions within Ghana and to other countries, as our sample was representative of the student population in the Eastern Region only. Again, our outcome is based on self-reports, which are prone to social desirability bias; knowing that the study was about blood donation, it is possible that some students over-reported their willingness to donate blood. The high response rate recorded (95.9%) may be attributed to the direct administration of questionnaires to students, as no incentive was given. However, the schools assigned a coordinator who introduced the survey team to the students. This approach may have influenced students' participation in the study. Students may have felt obliged to complete the questionnaire or provide favourable responses, despite the study being clearly explained as voluntary.

Given these limitations, we view this study as a starting point in ways to improve blood donation by a young, healthy population in Ghana.

## 5. Conclusion

In conclusion, unwillingness to VDB is prevalent among students in the Eastern Region of Ghana. This is shaped by a constellation of cognitive (knowledge, perceived eligibility, safety beliefs), sociocultural (normative discouragement), and structural (institutional and logistical) factors. Enhanced education of students on BD processes, eligibility and safety is highly recommended. Education may be facilitated through the creation of health clubs in schools and the incorporation of BD awareness into student orientation programmes or curriculum. Organisation of free medical checkups at least once a year for students, and the development of guidelines to allay fears of potential donors is recommended to enhance students' willingness to VDB.

## Supporting information

**S1 File. The data of the study.**
(XLS)

## Acknowledgments

The authors are grateful to all the respondents and the management of the schools that participated in the study.

## Author contributions

**Conceptualization:** Foster Amponsah-Manu, Augustine Badu, Augustine Kyere, Richmond Afrifa, Richmond Ampofo, Gloria Akofa Quarshie, Arko Akoto Ampaw.

**Formal analysis:** Augustine Badu, Yubraj Acharya, Arko Akoto Ampaw.

**Methodology:** Augustine Badu, Philomena Quayson.

**Project administration:** Augustine Badu, Augustine Kyere, Philomena Quayson, Samuel Attua, Richmond Afrifa, Eunice Serwaa Kegyeda, Samuel Morton, Richmond Ampofo, Gloria Akofa Quarshie, Offeibea-Effah Adarkwa.

**Resources:** Foster Amponsah-Manu, Arko Akoto Ampaw.

**Supervision:** Foster Amponsah-Manu, Augustine Badu, Jocelyn Asibey, Samuel Attua, Arko Akoto Ampaw.

**Writing – original draft:** Foster Amponsah-Manu, Augustine Badu, Yubraj Acharya, Jocelyn Asibey, Eunice Serwaa Kegyeda, Samuel Morton, Offeibea-Effah Adarkwa, Arko Akoto Ampaw.

**Writing – review & editing:** Foster Amponsah-Manu, Augustine Badu, Yubraj Acharya, Arko Akoto Ampaw.

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
