## [Decision Letter · Decision Letter 0]

23 Jan 2026

PONE-D-25-60625Factors associated with low voluntary blood donation turnout of students in the Eastern Region, GhanaPLOS One

Dear Dr. Badu,

Thank you for submitting your manuscript to PLOS ONE. After careful consideration, we feel that it has merit but does not fully meet PLOS ONE’s publication criteria as it currently stands. Therefore, we invite you to submit a revised version of the manuscript that addresses the points raised during the review process.

The manuscript has been evaluated by two reviewers, and their comments are available below and in the attached document. The reviewers have raised a number of major concerns.

Could you please carefully revise the manuscript to address all comments raised?

We look forward to receiving your revised manuscript.

Kind regards,

Ilse Bloom

Staff Editor

PLOS One

Journal Requirements:

2. We note that Figure 1 in your submission contain [map/satellite] images which may be copyrighted. All PLOS content is published under the Creative Commons Attribution License (CC BY 4.0), which means that the manuscript, images, and Supporting Information files will be freely available online, and any third party is permitted to access, download, copy, distribute, and use these materials in any way, even commercially, with proper attribution. For these reasons, we cannot publish previously copyrighted maps or satellite images created using proprietary data, such as Google software (Google Maps, Street View, and Earth). For more information, see our copyright guidelines: http://journals.plos.org/plosone/s/licenses-and-copyright.

a. You may seek permission from the original copyright holder of Figure(s) [#] to publish the content specifically under the CC BY 4.0 license.

3. We note that there is identifying data in the Supporting Information file <data-blood donation.xls>. Due to the inclusion of these potentially identifying data, we have removed this file from your file inventory. Prior to sharing human research participant data, authors should consult with an ethics committee to ensure data are shared in accordance with participant consent and all applicable local laws.

-Location data

Reviewers' comments:

Reviewer's Responses to Questions

**Comments to the Author**

1. Is the manuscript technically sound, and do the data support the conclusions?

Reviewer #1: Partly

Reviewer #2: Yes

2. Has the statistical analysis been performed appropriately and rigorously? 

Reviewer #1: No

Reviewer #2: No

3. Have the authors made all data underlying the findings in their manuscript fully available?

Reviewer #1: No

Reviewer #2: Yes

4. Is the manuscript presented in an intelligible fashion and written in standard English?

Reviewer #1: Yes

Reviewer #2: No

5. Review Comments to the Author

Reviewer #1: The study addresses an important public health issue and has the potential to make a meaningful contribution. However, substantial revisions are necessary to improve conceptual coherence, strengthen interpretation of analyses, and align the title, objectives, results, and conclusions.Such rewriting will make the manuscript clearer and of potential research significance.

Reviewer #2: Thank you for the opportunity to review this manuscript. Increasing blood donation turnout is critical for public health services. Students are often a key population, easily accessed through educational institutions. Understanding why donor presentation is low among students could help design effective interventions and improve blood drive yields. The authors address a clear gap in the literature and present a sound study design. The findings can help policymakers and blood services in Eastern Ghana decide where to direct their efforts most fruitfully. I have some suggestions on how the authors could improve the manuscript below. A fuller explanation about the multivariable statistical approach would also help readers interpret the findings. In several places the writing needs to be clarified to improve reader comprehension.

P2 Abstract. “About 40.3%”. Suggest deleting “About” if giving a precise percentage to one decimal place.

P2 Abstract .“Respondents were unwilling to donate without incentives …” is the denominator respondents who were unwilling to donate (n=887) or all respondents? What does the aOR refer to i.e. what is the referent or comparison group?

P2 Abstract. “Most preferred free health check-ups as an incentive”. The n=1,594 respondents, can the authors please explain which denominator this is?

P2 Abstract. “Thought of”. Consider replacing with “beliefs about” or similar.

P2. Abstract. Consider including the results from statistical analysis for the remaining factors mentioned (aOR etc).

Conclusion: “directives”, as the study is about voluntary blood donation, consider changing this word so it doesn’t imply coercion.

P3. “Out of the total of 187,280 blood donations collected in 2024” consider deleting the number as it repeats the sentence above.

P3. define “second-cycle” students?

P3. Can the authors provide more information about how “low” the student turnout is in Eastern Ghana? Is there a comparison with other regions, or has it declined over time? This would help strengthen the case for the study.

P4. Consider replacing “proper” with “accurate” knowledge

P4/5 “Thus, to address the challenge of low students’ turnout for voluntary blood donation programmes in the Eastern Region of Ghana, it was essential to investigate the specific barriers faced by the students in the region to inform the formulation and implementation of effective strategies to increase participation.” Consider editing this sentence down as it is quite long e.g. …“we investigated” … (and delete “specific” and “effective” as they aren’t necessary).

P5. Consider deleting the latitude and longitude data as it’s not clear these are relevant to the paper?

P5. Research design. Can the authors describe who led and conducted the study?

P6. Selection. “…from the 21 districts”. The previous page mentioned only 20 districts?

P6. “The proportion of second-cycle schools … balloting”. I found these descriptions of the sampling process confusing, could the authors please explain these more clearly?

Fig .2. Please confirm that Fig.2 doesn’t breach confidentiality agreements in the research ethics permissions i.e. is it necessary to name the schools that participated?

P6. The authors describe how schools were selected, but how were students at those schools selected to participate?

P7. Inclusion criteria. Add “written” informed consent?

P7 Data collection. “attitudes” towards blood donation? Please describe what “student orientation on blood donation” refers to? Can the authors replace “etc” with the factors in full? Was there a question on donation history? Consider deleting “Whereas”.

P8. Can the authors explain in more detail the approach used for multivariable statistical analysis? The Tables suggest that multiple analyses were undertaken, separating our different domains, rather than combining them?

P8. Ethics. Change “various schools” to “each school”?

P8. Consider changing “filled” to “completed” questionnaires, and “placed under lock and key in a secure location” to “stored securely”.

P9. Results. Can the authors describe the response rate of completions : students invited into the study, and any significant variations in the response rate?

P9. Table 1. Can the authors please provide the chi-square p-values as well as the test statistic (or instead of)? Include in the other tables as well.

P10. Consider changing the subheading to “Experiences and knowledge about blood donation …”

P10. Suggest delete only “about” 13.5%, as this is a precise proportion. Suggest adding “in future” at the end of the first sentence.

P10. Fig 3. Suggest Fig 3 is separated into two charts and the middle section deleted and mentioned in text only, as it only has two options. Each Fig also needs a vertical axis.

P10. Throughout the manuscript, suggest replacing “knowledge on” with “knowledge about” blood donation. Also suggest replacing “Thought of BD eligibility” with “Believe they are eligible to donate blood” or similar.

P11. Suggest adding “or” into the first sentence, and “also” in the second sentence.

P11. Can the authors please describe more fully in the Methods how “social stigma towards BD” and “Myths about BD” were measured?

P14. Discussion. Consider adding “students” in the first sentence. By convention, the first paragraph of a Discussion typically summarises the key points from the study as a whole, whereas this focuses on gender. Can the authors consider replacing it with a high-level synthesis of the key findings?

P16. Regarding the positive associations of peer and institutional support on blood donation willingness, what are some recommendations for policy and practice?

P17. What might a “holistic response” to logistical challenges look like?

6. PLOS authors have the option to publish the peer review history of their article (what does this mean?). If published, this will include your full peer review and any attached files.

Reviewer #1: **Yes:** Wilhellmuss Mauka

Reviewer #2: No

---

## [Author Response · Author response to Decision Letter 1]

7 Mar 2026

Response to general journal comments

1. Please, we have revised the manuscript to meet PLOS ONE's style requirements as much as possible

2. Please the previous Fig 1, which contained a map, has been taken off due to copyright issues.

3. Please the data set has been anonymised as much as possible. School names have been coded, and the programme of study column has been removed.

Response to reviewer one’s comments

Comments have been addressed in the manuscript and highlighted as red. These include:

1. Title and Conceptual Alignment

• The title is misleading. The analysis focuses on willingness or unwillingness to voluntarily donate blood (VBD) rather than actual low blood donation or low turnout. These concepts represent different stages of the blood donation pathway (intention vs behaviour)..

Please the title has been change to align with the analysis done: Factors associated with students’ unwillingness to voluntarily donate blood in the Eastern Region of Ghana.

• Authors should make explicit and clear their view on whether the primary problem is low blood donation, low turnout, non-voluntary donation, or unwillingness and should maintain consistency across the title, objecting, literature review, results, and conclusions.

The actual problem is low voluntary blood donation by students in the region. We have maintained consistency in the revised manuscript. The revision is highlighted in red.

• If unwillingness is conceptualised as a determinant of low blood donation, this should be explicitly stated and reflected in the title and analytical framing.

Please unwillingness is conceptualised as a determinant of low voluntary blood donation. This has been explicitly discussed in the background and highlighted in red. Page 4

2. Literature Review and Justification

• The literature review and justification are conceptually inconsistent. They alternate between factors influencing willingness, turnout, and low donation as if they are interchangeable.

Literature review and justification has been revised to be consistent with factors influencing willingness. Page 4, 5

• The justification is therefore double-barrelled and obscures the primary research problem.

Justification has been revised accordingly. Page 5

• The literature review should be reframed to focus on factors influencing low blood donation, with willingness clearly positioned as one explanatory construct if retained as the outcome.

Please literature has been revised accordingly. Pages 4, 5

3. Study Design and Methods Description

• The study includes inferential analyses; therefore, it should be described as a cross-sectional analytical study, not purely descriptive or simplt cross-sectional study.

This has been revised in Research design section at page 6

• Clarify whether the questionnaire was self-administered or interviewer-administered.

Please this is clarified in the data collection section at page 9: Respondents were handed the questionnaires to fill out after the purpose of the survey had been explained to them in plain language and had signed the informed consent form.

4. Sample Size and Sampling Procedures

• The manuscript does not explain how the student sample size (n = 2,467) was determined, nor whether this represented a minimum required sample size. A formal sample size calculation should be described.

Thank you for pointing this out. In the manuscript, we have added the following section to clarify our thinking process when we designed the study.

Statistical power. Our primary outcome is willingness to donate blood (binary; =1 if the respondent said they would be willing to donate blood). Although we intended to examine several key determinants, our sample size calculation was based on the student’s knowledge of blood donation, as this would provide a potential entry point for intervention (for example, if we found that students with limited knowledge were less likely to donate blood, the intervention would be educational programs to increase that knowledge). Based on data from National Blood Service Ghana, we assumed that approximately 30% students would be willing to donate blood. We also assumed that there would be three types of students: those with very little knowledge, those with some knowledge, and those with a lot of knowledge. They correspond to Low, Moderate, and High in “Knowledge of BD” measure in Table 2. We assumed that the proportion of students in these categories would be roughly 20%, 60%, and 20%. We also assumed about 10 percentage point difference in the probability of being willing to donate blood (equivalent to an odds ratio of about 1.55) between High and Low would be meaningful. Under these assumptions, a sample size of 1,920 students—or, approximately, 80 students each from the 24 schools—provides approximately 90% power (on a one-sided test with significance level of 0.05). When we visited the schools for data collection, we surveyed all eligible students since the marginal cost of collecting data from an additional student once we were there was low. Therefore, we ended up with 2,200 students who constitute our analytic sample.

• The sampling procedures lack clarity. The authors should:

o Clearly state the total number of second-cycle and tertiary schools in the Eastern Region at the time of the study.

Total number of schools in the Eastern Region at the time of the study is provided- page 7, school selection.

o Clarify whether “visited schools” represents all existing schools or only those previously reached by the blood service.

Please visited schools represent only those previously reached by the blood service. This has been clarified in the manuscript. Page 8, school selection

o Justify the selection (sampling) of 21 second-cycle and 3 tertiary schools, particularly given that some districts reportedly had no schools and that only five tertiary institutions exist.

Out of the total of 74 second-cycle and 31 tertiary schools in the Eastern Region, the Blood Donor Unit of the Eastern Regional Hospital visited 59 second-cycle schools in 21 districts for voluntary blood donation drives in 2024.

A proportionate stratified sampling technique was adopted to select 21 second-cycle schools out of the 59 visited. Based on the proportion, districts with 7, 5, 4, 3, 2 and 1 schools had 2, 2, 1, 1, 1 and 0 schools selected from them respectively. This resulted in 20 schools, so a school from one of the districts with one school and zero selection was picked randomly and added to the 20 schools, making 21.

Also, the Donor Unit visited 5 tertiary schools in 5 different districts for voluntary donation drives in 2024. Three tertiary schools were randomly selected out of the 5 visited, for the study.

Please we relied on previously visited schools because we already had their donation records and good rapport with school authorities, so it was easier getting permission to conduct the study in those schools.

o Ensure the word selected does not mean sampled for methodological clarity.

We use selected for the schools because they were balloted from the districts

o Clearly describe how students were sampled within schools (e.g., sampling frame, systematic or simple random sampling).

Please how students were recruited has been clearly described under Sampling technique section at page 8

5. Statistical Analysis

• The study employed multistage sampling (districts, schools, students), yet the hierarchical structure is not adequately addressed statistically.

In the revised manuscript, we have clarified that we clustered the standard errors at the level of the school, thus allowing arbitrary correlation in the willingness to donate blood among students in a given school (Data analysis section. page 10). An alternative approach would have been to use a two-level mixed-effects logistic regression model with a random intercept for school to account for clustering of students within schools. However, since we are interested in broad determinants of willingness to donate blood than on obtaining precise estimate (i.e., with correct magnitude) of the association between each variable and the outcome, we have erred on the side of simplicity.

• The authors should clearly state the primary unit (cluster)—district or school.

The primary unit is the school. We did not randomly select the districts but selected the schools within them randomly.

• It should be clarified whether clustering effects at district or school level were assessed, including their contribution to variance.

Please see above. We cluster the standard errors at the school level.

• Describe how the regression model accounted for clustering and hierarchy (e.g., clustered standard errors, multilevel modelling), and justify the chosen approach.

Please see above.

• The correct term is multivariable analysis, not multivariate analysis, as only one outcome variable was analysed.

We have edited the manuscript throughout to say multivariable analysis instead of multivariate. Thank you for catching this!

• The manuscript should clearly explain how variables were selected for inclusion in the multivariable model (e.g., based on bivariate significance, theoretical relevance).

Significant bivariate variables were entered into a multivariable analysis. Clearly stated at Data analysis section- page 10

6. Results Presentation

• The Results section is incomplete. Although tables include both descriptive and multivariable findings, the text describes mainly descriptive results.

• The narrative should explicitly interpret adjusted results by stating:

o Whether an association exists

o The strength of association (adjusted odds ratios with 95% CIs)

o The direction of association

Please results section has been revised per the recommendation and highlighted in red. Page 11 to 16, results section

• Ensure consistency in reporting proportions (e.g., x% [n/N]) throughout the Results section.

Consistency in reporting proportions adhered to. Pages 11 to 16, results section

7. Discussion

• The Discussion is largely framed around statistical significance versus non-significance and does not adequately interpret the magnitude or direction of associations.

• The Discussion should focus on how specific factors increase or decrease willingness to donate blood.

Discussion has been reframed to focus on how specific factors increase or decrease willingness to donate blood.

Response to reviewer two’s comments

Comments have been addressed in the manuscript and highlighted as red. These include:

P2 Abstract. “About 40.3%”. Suggest deleting “About” if giving a precise percentage to one decimal place.

Please “About” deleted as recommended

P2 Abstract .“Respondents were unwilling to donate without incentives …” is the denominator respondents who were unwilling to donate (n=887) or all respondents? What does the aOR refer to i.e. what is the referent or comparison group?

Please we meant to say that lack of incentives was associated with unwillingness to donate (aOR 3.85, 95% CI: 1.85 – 8.02). The reference group is the ‘others’ (Table 4).

This has been clarified and revised in the Abstract

P2 Abstract. “Most preferred free health check-ups as an incentive”. The n=1,594 respondents, can the authors please explain which denominator this is?

We asked the question: ‘Would you be willing to donate if there were incentives?’. 1,594 respondents answered ‘Yes’. Out of those who answered ‘yes’, most preferred free health check-ups. Fig 3 has been revised to capture the frequencies. Please the ‘lack of incentives’ at the logistical challenges section was a different question.

Abstract has been revised accordingly.

P2 Abstract. “Thought of”. Consider replacing with “beliefs about” or similar.

Has been replaced with ‘perceived’

P2. Abstract. Consider including the results from statistical analysis for the remaining factors mentioned (aOR etc).

Results of other factors has been included

Conclusion: “directives”, as the study is about voluntary blood donation, consider changing this word so it doesn’t imply coercion.

Conclusion has been revised accordingly

P3. “Out of the total of 187,280 blood donations collected in 2024” consider deleting the number as it repeats the sentence above.

The sentence has been revised accordingly and highlighted in red

P3. define “second-cycle” students?

Second-cycle students have been defined as suggested.

P3. Can the authors provide more information about how “low” the student turnout is in Eastern Ghana? Is there a comparison with other regions, or has it declined over time? This would help strengthen the case for the study.

Please per the recommendations of reviewer 1, low turnout has been revised to low voluntary donation rate. This has run throughout the revised manuscript to maintain consistency.

Now on page 4

P4. Consider replacing “proper” with “accurate” knowledge

Revised accordingly

P4/5 “Thus, to address the challenge of low students’ turnout for voluntary blood donation programmes in the Eastern Region of Ghana, it was essential to investigate the specific barriers faced by the students in the region to inform the formulation and implementation of effective strategies to increase participation.” Consider editing this sentence down as it is quite long e.g. …“we investigated” … (and delete “specific” and “effective” as they aren’t necessary).

The sentence has been revised

P5. Consider deleting the latitude and longitude data as it’s not clear these are relevant to the paper?

Deleted as recommended

P5. Research design. Can the authors describe who led and conducted the study?

The study was conducted by the Blood Donation Committee of the Eastern Regional Hospital, led by the Clinical and Medical Directors.

This has been captured in the revised manuscript. Page 6.

P6. Selection. “…from the 21 districts”. The previous page mentioned only 20 districts?

The Eastern Region has a total of 74 second-cycle and 31 tertiary schools (18,19). Out of these, the Blood Donation Unit of the ERH visited 59 second-cycle schools in 15 districts and 5 municipalities, for voluntary blood donation drives in 2024.

Revision has been made. Page 8

P6. “The proportion of second-cycle schools … balloting”. I found these descriptions of the sampling process confusing, could the authors please explain these more clearly?

The description of the sampling procedure has been revised to improve clarity and conciseness. The section now clearly explains the proportional allocation of schools across districts and the subsequent selection of schools using simple random sampling (balloting).

Fig .2. Please confirm that Fig.2 doesn’t breach confidentiality agreements in the research ethics permissions i.e. is it necessary to name the schools that participated?

We have no permission from the schools to publish their names, so Fig 2 has been taken off.

P6. The authors describe how schools were selected, but how were students at those schools selected to participate?

Please how students were recruited has been added under the section- Student recruitment. Page 9

P7. Inclusion criteria. Add “written” informed consent?

Witten has been added to informed consent. Page 9

P7 Data collection. “attitudes” towards blood donation? Please describe what “student orientation on blood donation” refers to? Can the authors replace “etc” with the factors in full? Was there a question on donation history? Consider deleting “Whereas”.

Please section has been revised and highlighted in red. Page 9.

Please we assessed if student orientation programmes included education on blood donation.

This has been clarified. Page 9

P8. Can the authors explain in more detail the approach used for multivariable statistical analysis? The Tables suggest that multiple analyses were undertaken, separating our different domains, rather than combining them?

Please all the significant bivariate variables were entered into a multivariable analysis. We only separated the different categories for presentation purposes. We would have a very long

---

## [Decision Letter · Decision Letter 1]

31 Mar 2026

PONE-D-25-60625R1Factors associated with students’ unwillingness to voluntarily donate blood in the Eastern Region of Ghana.PLOS One

Dear Dr. Badu,

Thank you for submitting your manuscript to PLOS ONE. After careful consideration, we feel that it has merit but does not fully meet PLOS ONE’s publication criteria as it currently stands. Therefore, we invite you to submit a revised version of the manuscript that addresses the points raised during the review process.

We look forward to receiving your revised manuscript.

Kind regards,

Holger A. Rau

Academic Editor

PLOS One

Journal Requirements:

Reviewers' comments:

Reviewer's Responses to Questions

**Comments to the Author**

1. If the authors have adequately addressed your comments raised in a previous round of review and you feel that this manuscript is now acceptable for publication, you may indicate that here to bypass the “Comments to the Author” section, enter your conflict of interest statement in the “Confidential to Editor” section, and submit your "Accept" recommendation.

Reviewer #2: (No Response)

2. Is the manuscript technically sound, and do the data support the conclusions?

Reviewer #2: Yes

3. Has the statistical analysis been performed appropriately and rigorously? 

Reviewer #2: Yes

4. Have the authors made all data underlying the findings in their manuscript fully available?

Reviewer #2: Yes

5. Is the manuscript presented in an intelligible fashion and written in standard English?

Reviewer #2: Yes

6. Review Comments to the Author

Reviewer #2: Thank you to the authors for addressing the feedback that has now strengthened the manuscript. I have two further comments:

The new text on sample size and response rate is appreciated. However, it is currently situated in Methods, whereas conventionally this should be in Results (the first Result reported): “A total of 2,573 students from the 24 institutions were invited to participate in the study using systematic random sampling. Out of these, 2,467 students filled the questionnaire, yielding a response rate of 95.9%. However, 267 responses were excluded based on the study exclusion criteria, resulting in a final sample of 2,200 respondents.”

Relatedly, the 95.9% response rate for a voluntary survey is extraordinarily (some would say suspiciously) high. Can the authors please add a comment on this in the limitations section of the Discussion, noting whether there were any incentives (formal or otherwise), whether the authors feel this influenced participants’ responses, and if so, in what way and direction?

7. PLOS authors have the option to publish the peer review history of their article (what does this mean?). If published, this will include your full peer review and any attached files.

Reviewer #2: No

---

## [Author Response · Author response to Decision Letter 2]

6 Apr 2026

RESPONSE TO REVIEWERS COMMENTS

I write to respond to reviewers' comments on the manuscript: Factors associated with students’ unwillingness to voluntarily donate blood in the Eastern Region of Ghana. PONE-D-25-60625R1

Response to general journal comments

Please the reviewer comments did not include a recommendation to cite specific previously published works.

The reference list has been checked for correctness and completeness.

No retracted paper has been cited as of the time of submission of this response.

Response to reviewer 2 comments

The new text on sample size and response rate is appreciated. However, it is currently situated in Methods, whereas conventionally this should be in Results (the first Result reported): “A total of 2,573 students from the 24 institutions were invited to participate in the study using systematic random sampling. Out of these, 2,467 students filled the questionnaire, yielding a response rate of 95.9%. However, 267 responses were excluded based on the study exclusion criteria, resulting in a final sample of 2,200 respondents.”

Response: Please, the text has been moved to the result section. page 11

Relatedly, the 95.9% response rate for a voluntary survey is extraordinarily (some would say suspiciously) high. Can the authors please add a comment on this in the limitations section of the Discussion, noting whether there were any incentives (formal or otherwise), whether the authors feel this influenced participants’ responses, and if so, in what way and direction?

Response: Please, this has been addressed in the limitation section on page 21 and highlighted in red.

The high response rate recorded (95.9%) may be attributed to the direct administration of questionnaires to students, as no incentive was given. However, the schools assigned a coordinator who introduced the survey team to the students. This approach may have influenced students’ participation in the study. Students may have felt obliged to complete the questionnaire or provide favourable responses, despite the study being clearly explained as voluntary.

---

## [Editor Report · Decision Letter 2]

12 Apr 2026

Factors associated with students’ unwillingness to voluntarily donate blood in the Eastern Region of Ghana.

PONE-D-25-60625R2

Dear Dr. Badu,

We’re pleased to inform you that your manuscript has been judged scientifically suitable for publication and will be formally accepted for publication once it meets all outstanding technical requirements.

Kind regards,

Holger A. Rau

Academic Editor

PLOS One
---

## [Editor Report · Acceptance letter]

PONE-D-25-60625R2

PLOS One

Dear Dr. Badu,

I'm pleased to inform you that your manuscript has been deemed suitable for publication in PLOS One. Congratulations! Your manuscript is now being handed over to our production team.

Kind regards,

on behalf of

Prof. Dr. Holger A. Rau

Academic Editor

PLOS One